# Comparison of qPCR protocols for quantification of "*Candidatus Saccharibacteria*", belonging to the Candidate Phyla Radiation, suggests that 23S rRNA is a better target than 16S rRNA

Stella Papaleo[1], Riccardo Nodari[1], Lodovico Sterzi[1], Enza D'Auria[2], Camilla Cattaneo[3], Giorgia Bettoni[1], Clara Bonaiti[1], Ella Pagliarini[3], Gianvincenzo Zuccotti[1,2], Simona Panelli[1]*, Francesco Comandatore[1]*

1 Pediatric Clinical Research Center "Invernizzi", Department of Biomedical and Clinical Sciences, University of Milan, Milan, Italy, 2 Department of Pediatrics, Buzzi Children's Hospital, University of Milan, Milan, Italy, 3 Sensory & Consumer Science Lab (SCS_Lab), Department of Food, Environmental and Nutritional Sciences, University of Milan, Milan, Italy

* simona.panelli1@unimi.it (SP); francesco.comandatore@unimi.it (FC)

## Abstract

### Background

Candidate Phyla Radiation (CPR) is a large monophyletic group encompassing about 25% of bacterial diversity. Among CPR, "*Candidatus* Saccharibacteria" is one of the most clinically relevant phyla. Indeed, it is enriched in the oral microbiota of subjects suffering from immune-mediated disorders and it has been found to have immunomodulatory activities. For these reasons, it is crucial to have reliable methods to detect and quantify this bacterial lineage in human samples, including saliva.

### Methods and results

Four qPCR protocols for quantifying "*Ca.* Saccharibacteria" (one targeting the 23S rRNA gene and three the 16S) were tested and compared. The efficiency and coverage of these four protocols were evaluated *in silico* on large genomic datasets, and *in vitro* on salivary DNA samples, already characterized by amplicon sequencing on the V3-V4 regions of the 16S rRNA. *In silico* PCR analyses showed that all qPCR primers lose part of the "*Ca.* Saccharibacteria" genetic variability, even if the 23S qPCR primers matched more lineages than the 16S qPCR primers. *In vitro* qPCR experiments confirmed that all 16S-based protocols strongly underestimated "*Ca.* Saccharibacteria" in salivary DNA, while the 23S qPCR protocol gave quantifications more comparable to 16S amplicon sequencing.

### Conclusion

Overall, our results show that the 23S-based qPCR protocol is more precise than the 16S-based ones in quantifying "*Ca.* Saccharibacteria", although all protocols probably

**Data Availability Statement:** All relevant data are within the manuscript and its Supporting Information files.

**Funding:** This work was supported by a grant "Finanziamento Linea 2" from Università degli Studi di Milano, Dipartimento di Scienze Biomediche e Cliniche, to F.C. (project number 40225 PSR2021). St.P. acknowledges the support of the APC Central Fund of the University of Milano. There was no additional external funding received for this study. The funder had no role in study design, data collection and analysis, decision to publish, or preparation of the manuscript.

**Competing interests:** The authors have declared that no competing interests exist.

underestimate specific lineages. These results underline the current limits in quantifying "*Ca*. Saccharibacteria", highlighting the needs for novel experimental strategies or methods. Indeed, the underestimation of "*Ca*. Saccharibacteria" in clinical samples could hide its role in human health and in the development of immune-mediated diseases.

## Introduction

In the last decades, culture-independent molecular methods allowed the discovery of a large new group of bacteria from environments and human bodies, now referred to as Candidate Phyla Radiation (CPR) [1–4]. Currently, this monophyletic bacterial lineage includes more than 70 phyla [5, 6] and is still called "candidate" due to the lack of cultivated representatives, except for a few exceptions [7]. CPR population structure is currently poorly understood and the size of the CPR group is still debated. Recently, it has been estimated that it encompasses about 25% of the bacterial diversity [8].

CPR are small-sized bacteria (0.2–0.3 μm) with reduced genome size (usually < 1 Mb) [9] lacking important pathways, as those for amino acids and nucleotide biosynthesis [2]. Shotgun metagenomics highlighted that they have an unusual ribosome composition, missing some ubiquitous bacterial genes, such as uL1, bL9, and/or uL30. Furthermore, they have a peculiar 16S rRNA gene sequence with introns and indels [10]. The few successful cultivation attempts led to the discovery of unique lifestyles, with CPR colonizing the surface of other bacteria within the community, and living as epibionts with mutualistic/parasitic lifestyles [11, 12].

CPR phyla as "*Candidatus* Saccharibacteria" (formerly known as TM7), "*Candidatus* Absconditabacteria" (SR1) and "*Candidatus* Gracilibacteria" (GNO1) are now considered as part of the microbiota of human healthy oral tract, stomach and skin. Furthermore, either observational and experimental studies converged in suggesting their medical importance [6, 13].

Among these lineages, "*Ca*. Saccharibacteria" is the most studied. It has been reported to represent at least 3% of the human core oral microbiota and to be enriched in dysbiotic microbiomes during infection and inflammatory states of the oral mucosa (e.g., periodontitis and gingivitis) [13], and beyond (i.e., in Inflammatory Bowel Disease patients) [6]. These bacteria live as obligate epibionts (either mutualistic or parasitic), colonizing the surface of *Actinomycetota*, a phylum of bacteria usually present in human oral microbiota. The *Actinomycetota* host can belong to species with the potential to cause proinflammatory effects to the human counterpart. The epibiont can in turn modulate these inflammatory effects and have immunomodulatory activities itself on the human host [12, 14]. These effects have been studied on *Nanosynbacter lyticus* (previously, TM7x), the first lineage within "*Ca*. Saccharibacteria", and the first CPR, to be isolated in coculture with its host, *Actinomyces odontolyticus* (now *Schaalia odontolytica*) strain XH001 [12]. *S. odontolytica* has a strong pro-inflammatory effect by inducing Tumor Necrosis Factor Alpha (TNF-α) gene expression in macrophages. *N. lyticus* is able to suppress TNF-α expression and to prevent the detection of its host by human macrophages [12]. This anti-inflammatory effect of *N. lyticus*, as well as of other "*Ca*. Saccharibacteria" species isolated in coculture in the meanwhile, have been confirmed by subsequent functional studies [14].

Due to the growing awareness of its clinical relevance, it is important to have reliable methods to detect and quantify "*Ca*. Saccharibacteria" in human microbiota in various physiological and pathological conditions. This is a necessary premise for more focused taxonomic and functional studies, to clarify their population structure and role in maintaining the host's

health status. Unfortunately, given the peculiar characteristics of CPR bacteria, current molecular methods work poorly on them, or give biased pictures, especially regarding the estimate of relative abundances. As regards the amplicon sequencing, the most frequently used "universal" primers on the 16S gene display a low efficiency in amplifying CPR sequences [2, 15]. Recently, qPCR protocols targeting 16S or 23S rRNA genes have been designed for the quantification of "*Ca*. Saccharibacteria" in various environments [16, 17].

In this work, we evaluated four published qPCR protocols for "*Ca*. Saccharibacteria", three designed on 16S and one on 23S rRNA gene. An *in silico* analysis was firstly performed on sequences representative of the whole known taxonomic variability within "*Ca*. Saccharibacteria" [16]. qPCR experiments were then performed using salivary DNA samples from children suffering from food allergy and matched controls. These samples came from a previous study in which we characterized the oral microbiota of allergic vs control children using the V3-V4 16S rRNA amplicon sequencing [18]. In that work, we found that the oral microbiota of allergic children was enriched in "*Ca*. Saccharibacteria" (3.8% vs 2.5% in controls) and unclassified bacteria (9.2% vs 5.6% in controls). Here, we reevaluated the presence and relative abundance of "*Ca*. Saccharibacteria" in these samples, also in the light of the *in silico* analyses, to get more insights into the drawbacks and distortions associated with the currently available protocols for detecting, quantifying and classifying this emerging bacterial lineage.

## Materials and methods

### Ethics statement

This study analyzed a collection of human salivary DNA samples already characterized by 16S rRNA amplicon sequencing in a previous microbiota study [18]. The research was conducted according to the declaration of Helsinki, and all methods followed the relevant guidelines and regulations. The Ethics Committee of ASST-Fatebenefratelli-Sacco approved the study (Ref. n. 2021/ST/041). Privacy rights of subjects were carefully observed and authors did not have access to information that could identify individual participants. During the recruitment of subjects, parents signed a written informed consent to allow their child to participate in the study. Samples and data were first accessed for research purposes on 01/02/2022.

### Rationale and selection of the primer pairs used in this work

The aim of this work is to compare, by *in silico* and *in vitro* analyses, the efficiency of available qPCR protocols for the detection and quantification of "*Ca*. Saccharibacteria". In particular, we focused on four qPCR protocols, using the following primer pairs: TM7314F/TM7-910R [19, 20] (here called "16S_p1"), Sac1031F/Sac1218R [21] ("16S_p2"), TM7_16S_590F/TM7_16S_965R [22] ("16S_p3") and SacchariF/SacchariR (here called "23S") [17]. See S1 Table for details.

Protocols 16S_p1 and 16S_p2 were chosen based on Takenaka et al. (2018) [16] that evaluated different primers for "*Ca*. Saccharibacteria" quantification. These authors concluded that the TM7314F/TM7-910R pair (16S_p1) gave the most reliable real time quantification, and for this reason we included it in our collection. The other pair, Sac1031-F/Sac1218R (16S_p2) in their hands appeared to underestimate their environmental samples, but because it was originally designed to analyze "*Ca*. Saccharibacteria" in mammalian feces [21] we decided to test it on our dataset. The third 16S pair, 16S_p3, described in [22], has already been recognized for its high coverage and specificity for "*Ca*. Saccharibacteria" [16].

The 23S protocol was chosen because primers were designed on the basis of a very recent genomic analysis [17] and because it targets a gene other than the 16S rRNA, which is known to have a limited capacity to detect CPR.

To evaluate the reliability of the "*Ca*. Saccharibacteria" quantifications obtained from the four qPCR protocols listed above (16S_p1, 16S_p2, 16S_p3 and 23S), *in silico* and *in vitro* PCR experiments were conducted. The *in vitro* experiments were carried out on a collection of salivary DNA samples previously characterized by D'Auria et al. (2023) [18], through 16S amplicon sequencing using the pro314F/pro805R primer pair [23] (from here called "16s_meta"). The aim was to compare the "*Ca*. Saccharibacteria" frequencies estimated from the qPCR protocols to those obtained from 16S amplicon sequencing. Unfortunately, qPCR protocols return an absolute quantification of "*Ca*. Saccharibacteria" which is not directly comparable to the frequencies obtained by D'Auria et al. (2023). Indeed, to estimate the "*Ca*. Saccharibacteria" frequency we need to determine also the total amount of bacteria in the analyzed sample. Thus, for each of the four qPCR protocols, we estimated the "*Ca*. Saccharibacteria" frequency as the ratio between "*Ca*. Saccharibacteria" quantification and the total amount of bacteria, determined using the qPCR pan-bacterial primers 926F/1062R (from here called "16S_panbacteria") [24]. See S1 Table for details.

### *In silico* PCR experiments

*In silico* PCR experiments were carried out on the six primer pairs listed above (16S_p1, 16S_p2, 16S_p3, 23S, 16S_panbacteria and 16s_meta). The primer pairs were *in silico* tested on two large datasets: a collection of "*Ca*. Saccharibacteria" sequences from the SILVA database and a collection of high quality "*Ca*. Saccharibacteria" genomes.

Regarding the SILVA database, the reference datasets LSU Ref NR99 v.138.1 (Large Subunit, i.e. the 23S rRNA gene) and SSU Ref NR99 v.138.1 (Small Subunit, i.e. the16S rRNA gene) were retrieved and the sequences annotated as "Saccharimonadia" (the only SILVA annotation relative to "*Ca*. Saccharibacteria") were extracted. Unfortunately, only two "Saccharimonadia" sequences were present in the LSU Ref NR99 v.138.1 (i.e. the 23S rRNA gene), and thus the *in silico* PCR analyses could be carried out only for the 16S-based protocols on the SSU Ref NR99 v.138.1 dataset (the 16S rRNA gene), from which 2,978 "Saccharimonadia" sequences were extracted. The *in silico* PCR analyses were performed using the ThermonucleotideBLAST tool [25] setting the following parameters:—primer-clamp 5— max-mismatch 6—best-match -m 1. The 2,978 extracted sequences were then aligned using the MAFFT tool [25] and phylogenetic analysis carried out using FastTree [26]. The results of the *in silico* PCR were mapped on the obtained phylogenetic tree using iTOL web tool [27].

The *in silico* PCR analysis was also carried out on the 16S rRNA and 23S rRNA gene sequences retrieved from a large manually curated collection of "*Ca*. Saccharibacteria" genomes, as follows. All the "*Ca*. Saccharibacteria" genome assemblies present into the BV-BRC database [28] as of June 27, 2023 were retrieved and subjected to 16S rRNA and 23S rRNA gene calling using Barrnap (github.com/tseemann/barrnap). The 16S rRNA sequences sized between 1,300 and 1,500 nt, and the 23S rRNA sequences sized between 3,000 and 3,500 nt, were considered complete. The genome assemblies harboring at least one complete 16S rRNA and one complete 23S rRNA genes were selected. For each genome, all the 16S rRNA gene sequences called by Barrnap were analyzed by *in silico* PCR as described above, using the five primer pairs targeting 16S (16S_p1, 16S_p2, 16S_p3, 16S_panbacteria and 16S_meta primers); the same was done for the 23S rRNA gene and the corresponding primer pair. The longest 16S rRNA sequence of each selected genome was extracted and subjected to phylogenetic analysis using FastTree, after alignment using MAFFT. The results of the six *in silico* PCR experiments (five on 16S rRNA gene target and one on the 23S rRNA gene) were mapped on the obtained phylogenetic tree using iTOL [27].

## Phylogenetic placement of the D'Auria et. al 2023 "*Ca*. Saccharibacteria" sequences

The V3-V4 16S rRNA sequences annotated as "*Ca*. Saccharibacteria" by D'Auria and colleagues (2023) [18] were retrieved and BlastN searched against both the two 16S rRNA datasets (from SILVA and genome assemblies) already used for phylogenetic analyses. For each sequence, the most similar sequence was highlighted on the phylogenetic trees using iTOL [26, 27].

### *In vitro* experiments

The four qPCR protocols for the quantification of "*Ca*. Saccharibacteria" were tested *in vitro* on 61 DNA samples previously characterized through 16S amplicon sequencing by D'Auria et al. [18]. In that study, DNA was extracted from saliva of patients suffering from food allergies and matched controls, and subjected to 16S amplicon sequencing. The same DNA tubes were used in this study: samples were not re-extracted in order to avoid any kind of variation that would have distorted the comparison between the qPCR results and the amplicon sequencing analysis. As stated above, the quantifications obtained for each of the four qPCR protocols included in the study (16S_p1, 16S_p2, 16S_p3 and 23S) were normalized on the total bacterial amount of the sample, estimated using the pan-bacterial primers 926F/1062R (here called "16S_panbacteria") [24]. Details are described below.

**End-point PCR.** For each of the protocols, a standard end-point PCR protocol was first run to verify specificity and provide amplicons for the standard curve for subsequent qPCR experiments. PCR reactions were performed on those salivary DNA samples that, following 16S rRNA amplicon sequencing, displayed the highest relative abundances of "*Ca*. Saccharibacteria". Amplifications were set up in a total volume of 20 μL containing: 10 μL GoTaq® Green Master Mix (Promega Corporation, Madison, Wisconsin, USA), 1 μL of each 10 μM primer, 6μL Promega PCR amplification-grade water (Promega) and 2 μL of the sample DNA (corresponding to about 20 ng). Cycling programs were performed on a Biorad T100 thermal cycler. Thermal profiles are listed in S2 Table. PCR products were analyzed through electrophoresis on 1% agarose gels. Amplicons were gel-purified using the Wizard® SV Gel and PCR Clean-Up System (Promega) and quantified with a Qubit 4 Fluorometer (Thermofisher scientific, Waltham, Massachusetts). DNA was finally diluted in Milli-Q water. Ten-fold serial dilutions were prepared for each amplicon that contained known numbers of fragment copies ranging from $10^7$ to 10 copies/μL to create the standard curves.

*Quantitative PCR.* Each 15 μL reaction contained 7,5 μL of 2x SsoAdvanced Universal SYBR® Green Supermix (BioRad, Hercules, California), 0,4 μL of each 10 μM primer, 4,7 μL of PCR amplification-grade water (Promega Corporation, Wisconsin, USA) and 2 μL of sample DNA (about 20 ng). Each sample was qPCR-amplified in three technical replicates. The qPCR assays were performed on a BioRad CFX Connect real-time PCR System (BioRad, Hercules). Thermal profiles are listed in S3 Table. The specificity of each primer pair was assessed through the melting profile generated at the end of each qPCR experiment, with a range of temperature between 60° and 95°C.

### Statistical analyses

The detecting capability of the four primer sets tested in this study (16S_p1, 16S_p2, 16S_p3 and 23S) was compared on the basis of the "*Ca*. Saccharibacteria" quantification provided by each of them, as follows. For each primer set, the "*Ca*. Saccharibacteria" representation in the total bacterial community was calculated, in percentage, as the ratio between their absolute

quantification and the pan-bacterial absolute quantification obtained using the 16S_panbacterial primers (see S1–S3 Tables for details). Results obtained in this way for each of the four qPCR primer sets, and those obtained in the 16S amplicon sequencing [18], were then compared with Mann-Whitney U test and linear regression (significant p value threshold 0.05), using R. For each of these five methods of quantification, the "*Ca*. Saccharibacteria" frequencies obtained for allergic vs control subjects were compared using Mann-Whitney U test, using R.

## Sequencing and analysis of 23S rRNA gene amplicon

Twelve representative samples selected from the 61 tested first by 16S amplicon sequencing (D'Auria et al., 2023) and then by qPCR were chosen for 23S amplicon sequencing, to verify the specificity of the primers and define the portion of the taxonomic variability of "*Ca*. Saccharibacteria" covered by these primers. Eight samples were chosen because they displayed the highest differences between the quantifications provided by the 23S qPCR and those obtained from the 16S amplicon sequencing, while other four samples were sequenced as controls. Sequences were performed on an Illumina Novaseq 6000 platform by MrDNA, Shallowater, Texas. Reads quality was assessed using the FastQC tool (http://www.bioinformatics.babraham.ac.uk/projects/fastqc). Then, the 23S rRNA gene amplicon reads were taxonomically assigned using the Mothur tool [29] and SILVA138.1 LSURef NR99 as reference database [30]. Briefly, reads were aligned against the reference Silva database and those containing chimeric information were removed. The remaining reads were grouped into Operative Taxonomic Units (OTUs) using the 0.05 distance threshold (without *a priori* information, the threshold has been determined on the basis of the nucleotide distance distribution). Then, a phylogenetic-based taxonomic annotation of OTUs was performed on the representative reads of the different OTUs. The reads were BlastN-searched against the NCBI nt database and, for the 20 best hits, sequences and taxonomic metadata were retrieved. The obtained NCBI sequences and the representative OTU sequences were aligned and subjected to Maximum Likelihood (ML) phylogenetic analysis using RAxML8 [31], with 100 pseudo bootstraps, using the model K80+G, as determined by best model selection analysis using ModelTest-NG [32].

## Results and discussion

### *In silico* PCR experiments

*In silico* PCR analyses were performed on sequences representative of the whole known taxonomic variability within "*Ca*. Saccharibacteria", retrieved from two large datasets. These sequences are the 2,978 16S rRNA annotated as Saccharimonadia retrieved in the SILVA database [30], and the 16S/23S rRNA sequences from a manually curated 114 "*Ca*. Saccharibacteria" genomes dataset (S4 Table).

Fig 1 shows the 16S rRNA-based phylogenetic trees obtained for the two datasets (hereafter referred to as "SILVA" and "genomes"), annotated with the results of the *in silico* PCR analyses for all the six sets of primers considered in this study (see Methods and S1 Table for details). The colored rings in Fig 1 indicate the taxonomic variability within "*Ca*. Saccharibacteria" successfully amplified by each pair. Results for SILVA (Fig 1A) evidenced that none of the protocols completely covered the taxonomic variability. As also shown in Table 1, the highest coverage was obtained for 16S_meta, i.e., the primers for 16S amplicon sequencing, that *in silico* amplified 98% (2,903) of the 2,978 "Saccharimonadia" 16S rRNA sequences in SILVA. Similarly, 97% of the sequences (2,875) was amplified by 16S_panbacteria primers, followed by 83% for 16S_p3 (2,482), 64% (1,908) for 16S_p1, and only 6% (168) for 16S_p2. As explained above (see Materials and Methods) this analysis could not include the 23S primers because of

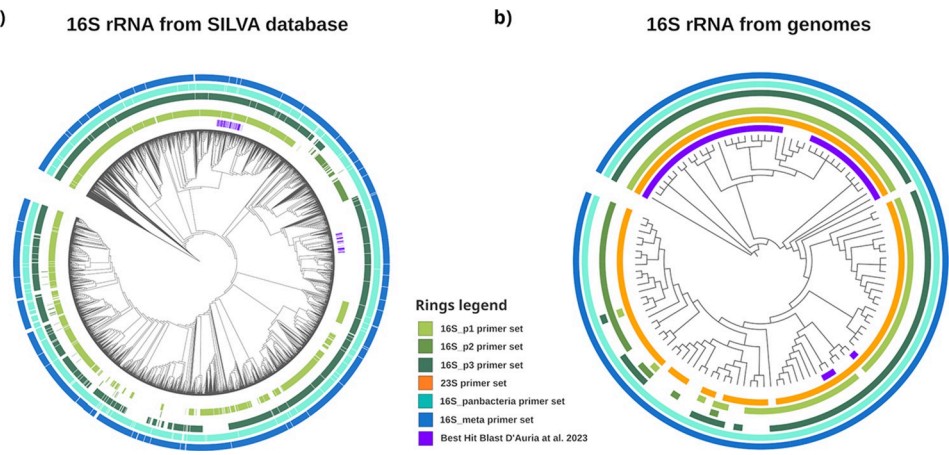

**Fig 1. *In silico* PCR amplifications mapped on "*Candidatus* Saccharibacteria" 16S rRNA phylogenetic trees.**
Results of in silico PCR amplifications were mapped on 16S rRNA phylogenetic trees to visualize the portions of "*Ca.* Saccharibacteria" known taxonomic diversity matched by the analyzed primer pairs. (a) Maximum Likelihood (ML) phylogenetic tree obtained from the 2,978 16S rRNA sequences annotated as "Saccharimonadia" in the 16S rRNA SILVA database. The five outer circles (coloured in a gradient from blue to green) represent the results of the *in silico* PCRs mapped on the tree, while the inner violet circle indicates the position of the lineages sequenced by D'auria et al. (2023). (b) ML phylogenetic tree obtained from 16S rRNA gene sequences retrieved from a manually curated collection of "*Ca.* Saccharibacteria" genomes. Similarly to above, the six outer circles (coloured from blue to yellow) map the results of the in silico PCRs experiments on the tree, while the violet inner circle maps the position of the best hits for the lineages sequenced by D'auria et al. (2023).

the poor representation of 23S rRNA sequences belonging to "*Ca*. Saccharibacteria" in the SILVA database. Overall, the results for SILVA showed that, in this quite large dataset, all *in silico* amplifications missed a variable portion of the currently known taxonomic variability within "*Ca*. Saccharibacteria" (probably far from exhaustive), with the best "performance" highlighted for the 16S_meta pair, which missed the 2% of Saccharimonadia 16S rRNA sequences. This suggests that, even though 16S_meta primers have a very high coverage for the "*Ca*. Saccharibacteria" phylum (which is the one, within the CPR group, for which most sequence data are available) they may conceivably fail to detect larger portions of the CPR taxonomic variability outside of "*Ca*. Saccharibacteria", thus leading to a possible underestimation of some phyla and the loss of information in studies relying on 16S rRNA amplicon sequencing. Indeed, a recent systematic survey analyzed the sequences from over 6,000 assembled metagenomes and evaluated 16S rRNA primers commonly used in amplicon studies. The authors observed that >70% of the bacterial clades systematically under-represented or missed in amplicon-based studies belong to CPR [15].

**Table 1. *In silico* analyses of the tested primer pairs.**

| Protocols | Coverage on 16S rRNA sequences of the SILVA Database | Coverage on 16S/23S rRNA sequences of "*Ca.* Saccharibacteria" genome dataset | Level of correlation with V3-V4 16S rRNA sequencing |
|---|---|---|---|
| 16S_p1 | 64% | 73% | highly correlated |
| 16S_p2 | 6% | 19% | low correlated |
| 16S_p3 | 83% | 75% | |
| 23S | / | 96% | highly correlated |
| 16S_meta | 97% | 100% | |
| 16S_panbacteria | 97% | 100% | |

Fig 1B shows the same analyses performed on the database of the 114 "*Ca*. Saccharibacteria" genomes. From Fig 1B it emerges that, once again, the pan-bacterial primer sets (16S_meta and 16S_panbacteria) are the most comprehensive, with a coverage of 100% (114 sequences). Among qPCR protocols, 23S was found to cover a greater portion of variability than those based on 16S. It successfully amplified 96% (109) of the sequences within the "*Ca*. Saccharibacteria" genome database, followed by 75% (86) amplified by 16S_p3 primer set, 73% (83) by 16S_p1, and 19% (22) by 16S_p2 (Table 1). The low coverage of the 16S protocols could be attributed to the peculiar sequence and structure of the 16S rRNA gene in members of Candidate Phyla Radiation. Indeed, as stated above, it presents introns, insertions and deletions that could be an obstacle for amplification [10].

Fig 1 also maps the position, on the two phylogenetic trees, of the best hits observed for the "*Ca*. Saccharibacteria" V3-V4 16S sequences obtained by D'Auria et al. (2023) [18]. It is interesting to note that none of the sequences obtained in this paper presented a perfect match with those deposited in the two datasets. In other words, both the SILVA and genomic datasets lacked sequences whose V3-V4 portions of 16S gene were identical to those sequenced by D'auria and colleagues in their dataset, showing that the "*Candidatus* Saccharibacteria" lineages expanded in allergic children could belong to an unexplored portion within the phylum.

### *In vitro* experiments

The next step was to experimentally evaluate the efficiency of the selected qPCR protocols (three based on the 16S and one on the 23S rRNA gene, S1 Table) on the collection of salivary DNA previously characterized by 16S amplicon sequencing [18]. In that paper, the authors found that the saliva of children suffering from food allergy, compared to matched controls, was enriched in "*Ca*. Saccharibacteria" and in sequences unresolved by the 16S amplicon sequencing. Phylogenetic analysis revealed that these bacteria belong to the CPR phylum "*Candidatus* Gracilibacteria".

For each protocol and for each sample, the representation of "*Ca*. Saccharibacteria" within the bacterial community was estimated as the ratio between the "*Ca*. Saccharibacteria" quantification obtained with the specific primer pair (16S_p1, 16S_p2, 16S_p3 and 23S) and the total bacterial estimate obtained with the universal pan-bacterial primer set 16S_panbacteria (S5 Table). These data were then compared to the relative abundances previously obtained by the 16S amplicon sequencing. The results of the comparisons are shown in Fig 2 and summarized in Table 1 (third column). Fig 2 shows that the quantifications obtained from three out of the four protocols (23S, 16S_p1 and 16S_p2) were significantly correlated to those obtained by 16S amplicon sequencing (linear regression, pvalue < 0.05) (Fig 2A–2C). Among these protocols, only the one based on the 23S rRNA gene produced estimates comparable to the 16S amplicon sequencing, both in terms of correlation and absolute quantification. Indeed, this protocol produced abundances not statistically different from 16S amplicon sequencing (Mann Whitney U test, pvalue > 0.05) (Fig 2E).

Instead, all the three qPCR protocols targeting the 16S rRNA gene underestimated the presence of "*Ca*. Saccharibacteria", both in the allergic and control groups. In fact, even though two of the 16S rRNA protocols were significantly correlated with the results of the 16S metagenomics (16S_p1 and 16S_p2, see Fig 2), the absolute quantifications provided for "*Candidatus* Saccharibacteria" differed from the 16S amplicon sequencing (and from the 23S protocol) by orders of magnitude.

Overall these results reflect the data of the *in silico* PCR conducted on the "*Ca*. Saccharibacteria" genome collections, confirming that, in *vitro* as well as in *silico*, the 23S protocol appears to be the most performing in terms of the portion of taxonomic diversity detected.

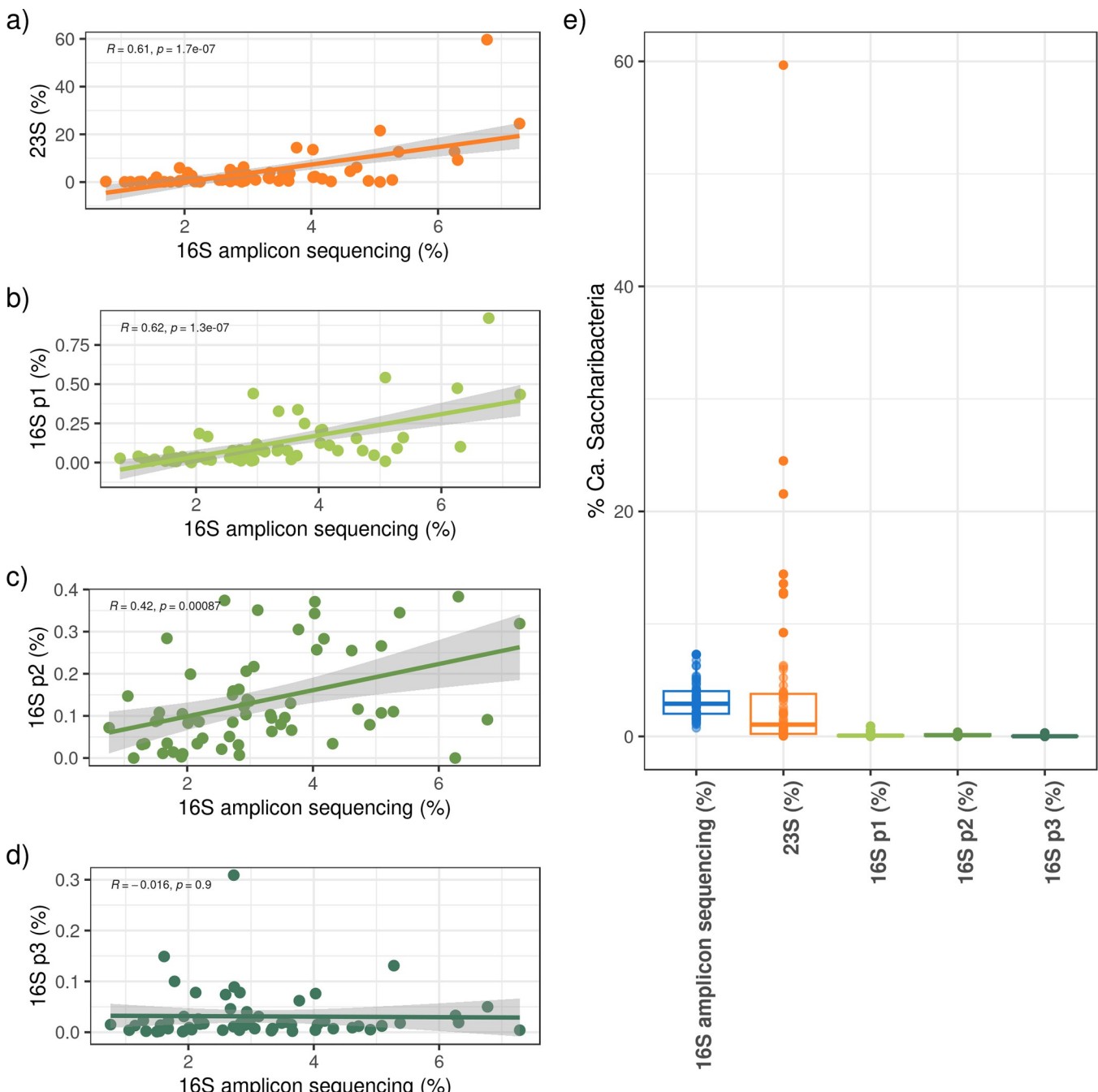

**Fig 2. Comparison between the "*Candidatus* Saccharibacteria" quantification obtained by 16S amplicon sequencing and the four tested qPCR protocols.** (a-d) Linear regression graphs of the "*Ca*. Saccharibacteria" quantifications obtained by: (a) 16S amplicon sequencing vs the 23S qPCR protocol (SacchariF-SacchariR); (b) 16S amplicon sequencing vs 16S p1 (TM7314F/TM7-910R); (c) 16S amplicon sequencing vs 16S p2 (Sac1031-F/Sac1218R); (d) 16S amplicon sequencing vs 16S p3 (TM7_16S_590F/TM7_16S_965R). For each plot, the Pearson correlation coefficients (Rs) and p-values are reported on the top and the confidence interval area is shown in gray. (e) Boxplot graph of the "*Ca*. Saccharibacteria" quantifications obtained by 16S amplicon sequencing and the four qPCR protocols. The median values are compared between 16S amplicon sequencing and the other four qPCR protocols by Wilcoxon test (p-values are reported on the plot).

Another point is that the relative abundance of "*Ca*. Saccharibacteria" provided by the 16S amplicon sequencing ranges between 0.8% and 7.3%, against a range of 0.04%-59.7% produced by the 23S protocol (see S5 Table). Thus, quantifications obtained from the 23S qPCR appear

to be scattered over a much broader range than those, more flattened, provided by the 16S amplicon sequencing. Overall, the differences between the 23S relative abundances and the 16S sequencing ones range between -5% and +52.9%. Interestingly, the two groups (controls and allergic subjects) significantly differ in terms of the distance between the quantification obtained by 23S qPCR and 16S amplicon sequencing. In controls, this difference ranges within a limited interval (from -2.5% to +10.7%) while in the allergic group it encompasses the whole interval (from -5% to +52.9%) (S1 Fig and S5 Table).

The difference between the two quantifications was > 5% in a total of seven subjects, five allergic patients and two controls (S1 Fig), thus highlighting the presence of a subset of samples, even if limited, for which the 23S qPCR protocol yielded a strongly higher quantification. For this reason, in order to exclude cross-reactions of the primers, and thus the amplification by qPCR of non-specific templates, we sequenced the 23S amplicons (see below).

Among the other protocols, the best performing 16S rRNA-based qPCR was the 16S_p1. The quantifications provided by this protocol correlated with those of the 16S amplicon sequencing but the absolute values were considerably lower. Therefore, they were not comparable in terms of absolute quantifications, clearly showing a strong underestimation of "*Ca*. Saccharibacteria".

There is one last important difference between the results obtained using qPCRs or 16S amplicon sequencing. This difference is related to the increase of lineages attributable to "*Ca*. Saccharibacteria" in allergic children. While the 16S amplicon sequencing returned a higher load of this phylum in allergic children compared to controls, these results were not confirmed by any of the tested qPCR protocols (Fig 3). This point shows very effectively how the choice to use a given technique over another can profoundly influence the final results and their interpretation in studies investigating these emerging CPR phyla and their role in the maintaining of the health status of the host. This limitation turns out to be particularly important in the case of groups such as "*Ca*. Saccharibacteria" whose role in immune-mediated diseases is increasingly evident.

## 23S rRNA qPCR amplicon sequence analysis

To exclude cross-reactions and contaminations in the 23S qPCR (see above), and have direct evidence on which "*Ca*. Saccharibacteria" lineages were amplified by this protocol (the first one to target a gene other than the 16S on "*Ca*. Saccharibacteria") amplicons from a selected subset of samples were sequenced on Illumina platform. A total of 940,756 sequences were produced and 819,506 (87,11%) of them passed the quality filtering steps. The analysis grouped these sequences into a total of 11 OTUs, of which the OTU1 contains 818,910 reads, corresponding to the 99.93% of the filtered reads (S6 Table). The S2 Fig shows a ML tree including the representative sequences of the 11 OTUs and their best hits retrieved from the NCBI nt database. The tree topology shows that nine out of 11 OTUs sequences (819,498 out of 819,506 sequences) cluster within "*Candidatus* Saccharibacteria". The remaining two OTU sequences (for a total of eight reads) are close to non-CPR bacteria.

These results excluded primers cross-reactions and the presence of non-specific amplicons. Therefore, the discrepancies observed with 16S amplicon sequencing, could be explained by hypothesizing the existence of "*Ca*. Saccharibacteria" lineages amplified by 23S and not by the primer pair "16S_meta" (i.e. 341F-805R, [23]). This point underlines the current lack of experimental approaches capable of detecting, in a comprehensive and reproducible way, the taxonomic diversity underlying "*Ca*. Saccharibacteria" and, probably even more so, all those CPR phyla for which sequence data are even scarcer.

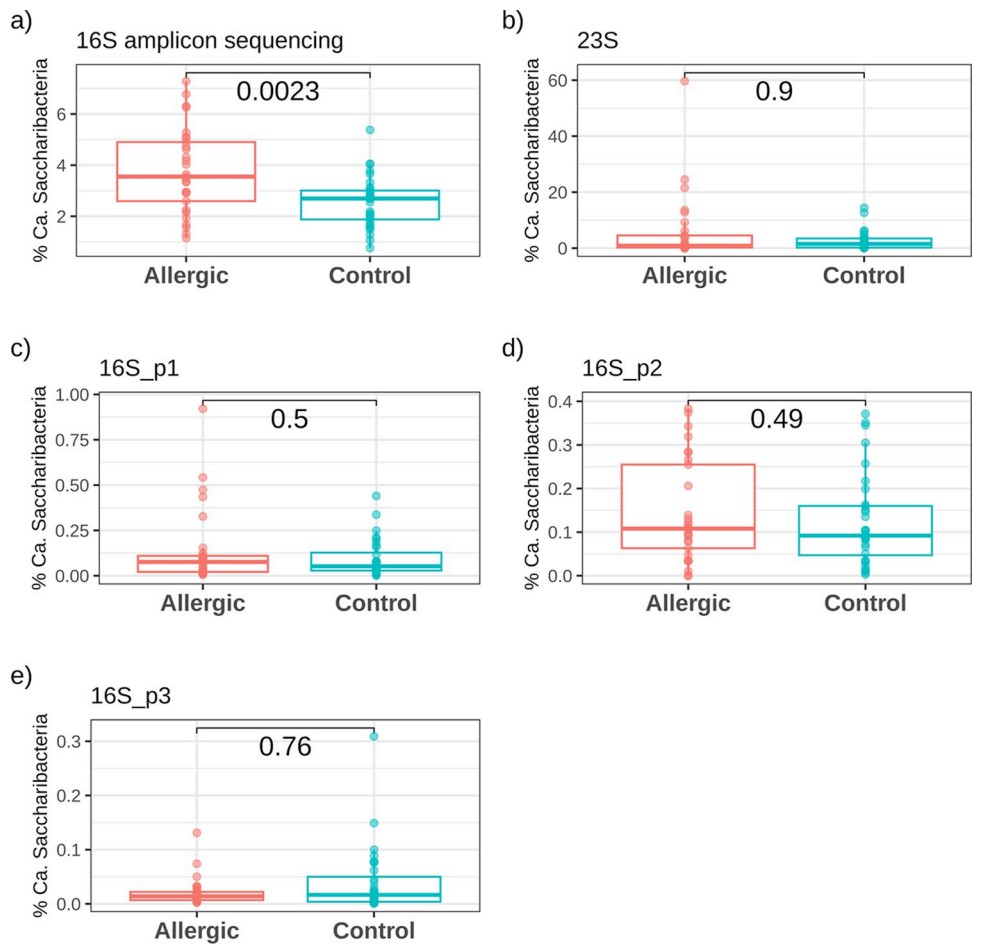

**Fig 3. Comparison of "*Ca*. Saccharibacteria" quantifications in allergic vs control patients obtained by 16S amplicon sequencing and the four tested qPCR protocols.** Boxplots reporting the quantifications obtained by: (a) 16S amplicon sequencing; (b) 23S qPCR protocol (SacchariF/SacchariR); (c) 16S_p1 qPCR protocol (TM7314F/TM7-910R); (d) 16S_p2 qPCR protocol (Sac1031-F/Sac1218R); (e) 16S_p3 qPCR protocol (TM7_16S_590F/TM7_16S_965R). Values obtained from allergic patients vs controls were compared using Wilcoxon test and the p-values are reported on the bars.

## Conclusions

Growing evidence currently highlights the importance of having a reliable method for the detection and quantification of Candidate Phyla Radiation (CPR) members in microbiota studies. Several papers have shown that 16S amplicon sequencing strongly underestimates CPR and that it is unable to efficiently resolve their taxonomy [2]. It has also been estimated that >70% of bacterial clades under-represented or missed in 16S amplicon sequencing studies belong to the CPR lineage [15]. This underestimation has several effects, particularly relevant when investigating immune-mediated diseases, considering that CPR lineages as "*Ca*. Saccharibacteria" have been experimentally observed to exert immunomodulatory roles in the human host and are enriched in several inflammatory conditions.

In recent years, several qPCR protocols targeting 16S or 23S rRNA genes have been designed for the quantification of "*Ca*. Saccharibacteria" in various environments. Four of these qPCR protocols were evaluated in this study, both *in silico* and in vitro, on large sequence databases or on saliva samples already characterized by 16S amplicon sequencing. Our results

show that none of the tested qPCR protocols is able to comprehensively and reproducibly detect the taxonomic diversity within "*Ca*. Saccharibacteria" and that each protocol likely introduces distortions in detection, quantification and reconstruction of taxonomic pictures. As for 16S amplicon sequencing, the regions sequenced in our previous study were the V3-V4. We cannot exclude that targeting other V regions (e.g., V1-V3) would have produced different results.

If this is the situation for the most investigated CPR phylum ("*Ca*. Saccharibacteria") it is reasonable to think that the limitations of currently available protocols will be much greater for other less studied CPR phyla.

It is becoming increasingly clear that this intriguing and ubiquitous part of the microbial world has important roles in clinical or environmental processes, and that these roles may have been greatly underestimated until now. To overcome these limitations, new experimental strategies are therefore necessary, such new amplification sequencing approaches based on new gene targets and/or workflows. These strategies should lead to more realistic pictures of CPR abundance within bacterial communities, providing more detailed taxonomic information and making it possible to investigate their fluctuations, associated with host inter-individual differences or pathogenic processes. These premises are necessary for more targeted and systematic functional studies, to clarify their role in maintaining the health status of the host and ecological roles in the environment.

## Supporting information

**S1 Fig. Distribution of the differences in "*Candidatus* Saccharibacteria" quantifications obtained by 23S qPCR and V3-V4 16S rRNA sequencing.** The two histograms report the distribution of the differences between the "*Candidatus* Saccharibacteria" quantifications obtained by 23S qPCR and by V3-V4 16S rRNA sequencing. Top: distribution of the differences for the allergic group. Bottom: distribution for controls. Dashed vertical lines indicate the interval between -5% and +5% difference between the 16S sequencing and 23S qPCR estimates.
(TIF)

**S2 Fig. Phylogenetic tree of the qPCR 23S amplicon sequences.** ML tree of the sequences representative of the OTUs obtained from the 23S amplicons and of background sequences retrieved from the nt NCBI database after BlastN search. "*Candidatus* Saccharibacteria" clade is coloured in red and the clade including non-CPR sequences is in gray. The number of sequences included in each OTU is reported on the leaves. Labels of the leaves from sequences retrieved from the NCBI nt database are omitted.
(TIF)

**S1 Table. Details of experimental conditions.** List of the primer pairs tested in this work.
(DOCX)

**S2 Table. Details of experimental conditions.** Thermal profiles used in the PCR experiments.
(DOCX)

**S3 Table. Details of experimental conditions.** Thermal profiles used in the qPCR experiments.
(DOCX)

**S4 Table. Cured dataset of 114 "Ca.** Saccharibacteria" genomes used for *in silico* PCR on 16S rRNA and 23S rRNA gene sequences.
(XLSX)

**S5 Table. "*Ca*. Saccharibacteria" quantification data by qPCR and 16S rRNA sequencing.**
(XLSX)

**S6 Table. Number of reads counted for each Operative Taxonomic Units (OTUs).**
(XLSX)

## Author Contributions

**Conceptualization:** Enza D'Auria, Simona Panelli, Francesco Comandatore.

**Data curation:** Riccardo Nodari, Lodovico Sterzi.

**Formal analysis:** Riccardo Nodari, Lodovico Sterzi, Francesco Comandatore.

**Funding acquisition:** Francesco Comandatore.

**Investigation:** Stella Papaleo, Camilla Cattaneo, Giorgia Bettoni, Clara Bonaiti, Simona Panelli.

**Supervision:** Simona Panelli, Francesco Comandatore.

**Writing – original draft:** Stella Papaleo, Riccardo Nodari.

**Writing – review & editing:** Stella Papaleo, Enza D'Auria, Camilla Cattaneo, Ella Pagliarini, Gianvincenzo Zuccotti, Simona Panelli.

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
