## [Decision Letter · Decision Letter 0]

5 Jun 2024

PONE-D-24-10132Unraveling and quantifying "Candidatus Saccharibacteria": in silico and experimental evaluation of V3-V4 16S rRNA metagenomics and qPCR protocolsPLOS ONE

Dear Dr. Panelli,

Thank you for submitting your manuscript to PLOS ONE. After careful consideration, we feel that it has merit but does not fully meet PLOS ONE’s publication criteria as it currently stands. Therefore, we invite you to submit a revised version of the manuscript that addresses the points raised during the review process. Please submit your revised manuscript by Jul 20 2024 11:59PM. If you will need more time than this to complete your revisions, please reply to this message or contact the journal office at plosone@plos.org. Please include the following items when submitting your revised manuscript:A rebuttal letter that responds to each point raised by the academic editor and reviewer(s). You should upload this letter as a separate file labeled 'Response to Reviewers'.A marked-up copy of your manuscript that highlights changes made to the original version. You should upload this as a separate file labeled 'Revised Manuscript with Track Changes'.An unmarked version of your revised paper without tracked changes. You should upload this as a separate file labeled 'Manuscript'.If applicable, we recommend that you deposit your laboratory protocols in protocols.io to enhance the reproducibility of your results. Protocols.io assigns your protocol its own identifier (DOI) so that it can be cited independently in the future. For instructions see: https://journals.plos.org/plosone/s/submission-guidelines#loc-laboratory-protocols. Additionally, PLOS ONE offers an option for publishing peer-reviewed Lab Protocol articles, which describe protocols hosted on protocols.io. Read more information on sharing protocols at https://plos.org/protocols?utm_medium=editorial-email&utm_source=authorletters&utm_campaign=protocols.

We look forward to receiving your revised manuscript.

Kind regards,

Guanglei Qiu

Academic Editor

PLOS ONE

Journal Requirements:

This work was supported by a grant "Finanziamento Linea 2" from Università degli Studi di Milano, Dipartimento di Scienze Biomediche e Cliniche to F.C. (project number 40225 PSR2021).

3. Please amend your authorship list in your manuscript file to include authors Dr. Giorgia Bettoni and Dr. Clara Bonaiti.

5. Please include your tables as part of your main manuscript and remove the individual files. Please note that supplementary tables (should remain/ be uploaded) as separate "supporting information" files.

Reviewers' comments:

Reviewer's Responses to Questions

**Comments to the Author**

1. Is the manuscript technically sound, and do the data support the conclusions?

Reviewer #1: Yes

Reviewer #2: Yes

Reviewer #3: Yes

2. Has the statistical analysis been performed appropriately and rigorously? 

Reviewer #1: Yes

Reviewer #2: Yes

Reviewer #3: Yes

3. Have the authors made all data underlying the findings in their manuscript fully available?

Reviewer #1: Yes

Reviewer #2: Yes

Reviewer #3: Yes

4. Is the manuscript presented in an intelligible fashion and written in standard English?

Reviewer #1: Yes

Reviewer #2: Yes

Reviewer #3: No

5. Review Comments to the Author

**Reviewer #1:** Reviewer Comments

This study focuses on the Candidate Phyla Radiation (CPR) organisms that representing a whopping 25% of bacterial diversity! Among them, the enigmatic Candidatus Saccharibacteria stands out, known for its immune-modulating prowess, especially in folks grappling with immune-related issues. They tested four different methods to detect and measure its presence, comparing their accuracy and efficiency. The results were illuminating: while traditional methods often missed the mark, a novel approach targeting the 23S rRNA gene emerged victorious, painting a clearer picture of Ca. Saccharibacteria's abundance. This is an important work, and must needed for the quickly expanding CPR and Saccharibacteria field. However, the manuscript can be improved following ways (see comments).

Major Comments

1. If these primers don’t work as well as the 16S metagenomics, what other methods can be used to test? Can you try some of those alternatives?

2. V3-V4 16S rRNA profiling does not generally considered metagenomics. Metagenomics provide MAG information, while 16S rRNA profiling generally provide abundance and prevalence only. “V3-V4 16S metagenomics” therefore is a misleading terminology.

3. How does comparing V1-V3 sequencing against these data and V3-V4 sequencing? V1-V3 sequencing generally doubt better for oral microbiome analysis and species level separation. eHOMD generally follow V1-V3 sequencing for oral microbiome analysis.

4. Authors picked Saccharibacteria specific 16S and 23S primer pairs from older studies. Authors also pointed out that these primers don’t cover all Saccharibacteria. Current availability of Saccharibacteria metagenome and genome have advanced nicely, and using these genome information to develop additional, either better 16S rRNA or 23S rRNA primers or core gene specific primers, will help advance the field and the quantification further. Is there a reason why this approach was not taken?

5. People with allergy may have only few Saccharibacteria species expanded so maybe it is worth testing species specific primers for those targeted species, rather than the whole genus?

Minor Comments

1. Not all CPR bacteria are cocci (line 73)

2. Amino acid vs aminoacid (line 74)

3. Figure 1 is very large in size compared to other figures

4. 23S primers also seem to detect other bacteria non-specifically, giving higher values compared to V3-V4 diversity.

5. Manuscript can use general proof reading.

**Reviewer #2: **Correct quantitative estimations are essential in microbial ecology, whereas in medical microbiology, their importance is difficult to overestimate. Modern breakthrough technologies are bringing closer to reality a situation where the procedure for analyzing microbiota based on 16S metagenomics will become a routine clinical analysis. From this point of view, the research done by the authors is relevant from the viewpoint of both fundamental and applied research. In general, the work has a consistent logical chain of experiments both in silico and in vitro, all methods are described in sufficient detail, and the results are presented either in the form of visualization of the processed data or in supplementary material. All comments that I can make on the manuscript being reviewed are of a technical nature. 

1. The authors have provided two list references on pp. 30-36 and 39-42. The manuscript contains references Ibrahim et al. 2021a (L. 69, 151) and Ibrahim et al. 2021b (L. 120, 161, 164), whereas the main list of references (L. 555-559) has only one. The second list of references contains references to (a) and (b), but they are the same (L. 741-745). Similar situation with references: Takenaka et al. 2018a (L. 120, 161, 164) and Takenaka et al. 2018b (L. 173).

2. L. 94: The correct name of the phylum Actinobacteria is Actinomycetota. Should be replaced here and further in the text. 

3. L. 309: Table S1 contains a list of primers, while the table, which is described in this paragraph L. 312-316 and further L. 334-340, has no name either in the text on page 43 or in a separate file designated file-4. It is necessary to determine the status of this table. In addition, the text presents data rounded to the nearest whole number, and the table shows the first decimal place. It is advisable to make a single presentation of the data, or at least correctly round to the nearest whole number. For example, L. 313-314: “in silico amplified 97.5% (2,903) of the 2,978 “Saccharimonadia'' 16S rRNA sequences in SILVA”, while the table shows 97%. 

4. L. 427: Table S6 is indicated, while in the table itself, Table S5 needs to be agreed upon.

**Reviewer #3: **The article's title does not depict the work. Since the qPCR targeting 23 S rDNA was better at resolving and quantifying the Ca, Saccharibacteria should also be depicted in the title.

Overall, this article has used and evaluated 03 16S rDNA bases and one 23S rDNA-based quantification of Ca. Saccharibacteria and correlated the percent abundance determined in a previous study by 16S metagenomic analysis. Furthermore, authors have sequenced the amplicons obtained by targeting 23S rDNA.

All the protocols and primers used are already reported and established.

The writing style is confusing and contradictory in places that need improvement.

The authors need to establish the article's novelty; This study reports nothing new except that all 4 methods were employed in one study.

Line 41-43. Rephrase. It is not clear

Line 43-48. There are contradictions in claims. Need clarity in writing.

Line 128-129. Give percent abundance of Ca. Saccharibacteria” and unclassified bacteria from the previous study based on 16 S rDNA metagenomics here.

Line 162. 16S rRNA is known to have a limited capacity to detect CPR already

Limitations of the study?

6. PLOS authors have the option to publish the peer review history of their article (what does this mean?). If published, this will include your full peer review and any attached files.

Reviewer #1: No

Reviewer #2: No

Reviewer #3: No

---

## [Author Response · Author response to Decision Letter 0]

10 Jul 2024

JOURNAL REQUIREMENTS:

The manuscript has been completely revised following PLOS ONE’s style requirements

This work was supported by a grant "Finanziamento Linea 2" from Università degli Studi di Milano, Dipartimento di Scienze Biomediche e Cliniche to F.C. (project number 40225 PSR2021).

Done. Please see the cover letter to the revised manuscript.

3. Please amend your authorship list in your manuscript file to include authors Dr. Giorgia Bettoni and Dr. Clara Bonaiti.

Done

The abstract has been revised to fulfil PLOS ONE’s rules and to answer specific points raised by one Reviewer. We have now paid attention that the abstract in the manuscript and the one uploaded on the online submission form are identical

5. Please include your tables as part of your main manuscript and remove the individual files. Please note that supplementary tables (should remain/ be uploaded) as separate "supporting information" files.

Done

Done

Please find below the changes to the reference list:

-We erroneously provided in the manuscript two lists of references. The correct list of references is the second one. On this, we have corrected further errors (see below)

-Ibrahim et al. 2021: it was cited two times due to an error. The double citation has now been removed

-same for Takenaka et al 2018. Again, this paper was erroneously cited twice. The double citation has now been corrected

REVIEWER #1: 

Major Comments

1. If these primers don’t work as well as the 16S metagenomics, what other methods can be used to test? Can you try some of those alternatives?

At the state of the art, the two main approaches for microbiota description are the 16S rRNA amplicon sequencing and the shotgun metagenomics. Of course, the latter is the best choice in terms of performance, but it is expensive, both in terms of economic and computational requirements. On the other hand, the amplicon sequencing approach provides good details about microbiota composition, requiring significantly less economic and informatic efforts, and for this reason it is the most frequently used. 

A different scenario opens up when the aim of the study is not to reconstruct the entire microbiota composition, but instead to focus on a specific taxon, or on a few taxa. In these cases, qPCR is reasonably the right choice: it is highly precise (indeed this approach is often used for absolute quantification, e.g. in RNA-Seq or 16S amplicon sequencing studies) and less expensive than NGS-based approaches.

Here, we address the problem of efficiently detecting and quantifying “Ca. Saccharibacteria” presence in the oral microbiota community, while studying alterations associated with an immune-mediated disease. We thus tested some of the (few) available qPCR protocols and compared their “performances” with the 16S rRNA amplicon sequencing, targeting the V3-V4 regions. A bioinformatics analysis accompanied the experimental work, to study the ability of these techniques to target the known taxonomic variability within “Ca. Saccharibacteria”. 

2. V3-V4 16S rRNA profiling does not generally considered metagenomics. Metagenomics provide MAG information, while 16S rRNA profiling generally provide abundance and prevalence only. “V3-V4 16S metagenomics” therefore is a misleading terminology.

We modified “V3-V4 16S rRNA metagenomics” to “V3-V4 16S rRNA amplicon sequencing” (or similar) throughout the manuscript.

3. How does comparing V1-V3 sequencing against these data and V3-V4 sequencing? V1-V3 sequencing generally doubt better for oral microbiome analysis and species level separation. eHOMD generally follow V1-V3 sequencing for oral microbiome analysis.

We agree with the referee: sequencing of other V regions could add information. Unfortunately, the analyses are unrepeatable, as for many of the samples the extracted DNA is finished. However, we added a sentence to comment on this point. See the section “Conclusions”. 

We would like to point out that the V3-V4 primer pair efficiently detects Bacteria as well as Archaea (Takahashi et al., 2014). Furthermore, recent work has shown that V3-V4 regions, although less used than others for oral microbiome studies, perform very well in this context because they allow an optimal OTU clustering at the 97% level, avoiding to cluster together sequences belonging to distinct genera. It has recently been observed that this is a problem that frequently affects several of the most frequently used primers pairs for the description of the oral microbiota (Reguera-Iglesias et al., 2023). 

Regueira-Iglesias A, Balsa-Castro C, Blanco-Pintos T, Tomás I. Critical review of 16S rRNA gene sequencing workflow in microbiome studies: From primer selection to advanced data analysis. Mol Oral Microbiol. 2023 Oct;38(5):347-399. doi: 10.1111/omi.12434. Epub 2023 Oct 7. PMID: 37804481.

Takahashi S, Tomita J, Nishioka K, Hisada T, Nishijima M. Development of a prokaryotic universal primer for simultaneous analysis of Bacteria and Archaea using next-generation sequencing. PLoS One. 2014 Aug 21;9(8):e105592. doi: 10.1371/journal.pone.0105592. PMID: 25144201; PMCID: PMC4140814.

4. Authors picked Saccharibacteria specific 16S and 23S primer pairs from older studies. Authors also pointed out that these primers don’t cover all Saccharibacteria. Current availability of Saccharibacteria metagenome and genome have advanced nicely, and using these genome information to develop additional, either better 16S rRNA or 23S rRNA primers or core gene specific primers, will help advance the field and the quantification further. Is there a reason why this approach was not taken?

We agree with the reviewer that the development of a new experimental methodology that more comprehensively detects and quantifies “Ca. Saccharibacteria” and other CPR bacteria would be highly desirable. This is an important goal and requires large experimental and bioinformatics efforts, which are indeed currently underway in our laboratory. However, the aim of the present work was not to develop new protocols, but to study the limitations of those already available. With this paper, we wish to address some of the (few) qPCR protocols already published for “Ca. Saccharibacteria”. We tested them and compared their “performance”, using combined in vitro and in silico approaches, on samples already analysed by 16S rRNA-based amplicon sequencing. We concluded that none of these available approaches yields a comprehensive and reproducible quantification and taxonomic picture of “Ca. Saccharibacteria”. 

5. People with allergy may have only few Saccharibacteria species expanded so maybe it is worth testing species specific primers for those targeted species, rather than the whole genus?

We completely agree with the reviewer. In fact, re-analysing the data from the D’Auria et al. 2023 paper, “Ca. Saccharibacteria” has been divided into at least 7 sub-lineages at taxonomic level of the genus. They are all low-abundance lineages, expanded in patients. Apart from TM7X and “Ca. Saccharimonas”, the others have incomplete taxonomic descriptions and consist of heterogeneous lineages. In any case, the aim of this work was to study the performance of the available protocols, which unfortunately target “Ca. Saccharibacteria” as a whole.

Minor Comments

1. Not all CPR bacteria are cocci (line 73)

Corrected

2. Amino acid vs aminoacid (line 74)

Corrected

3. Figure 1 is very large in size compared to other figures

We have tried to improve this

4. 23S primers also seem to detect other bacteria non-specifically, giving higher values compared to V3-V4 diversity.

To exclude cross-reactions and spurious amplifications, PCR products obtained with the 23S primers were sequenced on an Illumina Platform. As described in “Results and Discussion” (see “23S rRNA qPCR amplicon sequence analysis”), the analysis of the obtained reads showed that 819,498 reads out of a total of 819,506 (99,99%) clustered within “Ca. Saccharibacteria” (see supplementary Figure 3)

5. Manuscript can use general proof reading.

In revising it, we tried to improve the quality of the writing of the paper, also with the help of English-speaking colleagues. Moreover, we tried to improve the readability by simplifying and reorganising some parts, e.g. the abstract (also in response to specific requests by Reviewer #3), the Materials and Methods and some parts of Results and Discussion

REVIEWER #2: 

1. The authors have provided two list references on pp. 30-36 and 39-42. The manuscript contains references Ibrahim et al. 2021a (L. 69, 151) and Ibrahim et al. 2021b (L. 120, 161, 164), whereas the main list of references (L. 555-559) has only one. The second list of references contains references to (a) and (b), but they are the same (L. 741-745). Similar situation with references: Takenaka et al. 2018a (L. 120, 161, 164) and Takenaka et al. 2018b (L. 173).

We would like to apologise for providing two lists of references. It was an error that occurred using the Google Paperpile tool for formatting the references. We have now manually reviewed all the references. The correct list of references is the second one. On this, we have corrected the further errors signalled by the reviewers (same paper cited twice):

-for what concerns Ibrahim et al 2021: it was cited two times due to an error. The double citation has now been removed

-same for Takenaka et al 2018. It was erroneously cited twice. The double citation has now been removed

2. L. 94: The correct name of the phylum Actinobacteria is Actinomycetota. Should be replaced here and further in the text. 

Corrected

3. L. 309: Table S1 contains a list of primers, while the table, which is described in this paragraph L. 312-316 and further L. 334-340, has no name either in the text on page 43 or in a separate file designated file-4. It is necessary to determine the status of this table. In addition, the text presents data rounded to the nearest whole number, and the table shows the first decimal place. It is advisable to make a single presentation of the data, or at least correctly round to the nearest whole number. For example, L. 313-314: “in silico amplified 97.5% (2,903) of the 2,978 “Saccharimonadia'' 16S rRNA sequences in SILVA”, while the table shows 97%. 

We apologise for this error. The file designated “file-4” is in fact Table 1. It has now been correctly cited, and inserted at the end of the paragraph, as required by the PLOS ONE formatting style. The data presentation has been made consistent, with numbers rounded to the nearest whole number in the text, as suggested

4. L. 427: Table S6 is indicated, while in the table itself, Table S5 needs to be agreed upon.

Corrected

REVIEWER #3: 

1. The article's title does not depict the work. Since the qPCR targeting 23 S rDNA was better at resolving and quantifying the Ca, Saccharibacteria should also be depicted in the title.

We agree. The title has been changed as suggested. 

2. Overall, this article has used and evaluated 03 16S rDNA bases and one 23S rDNA-based quantification of Ca. Saccharibacteria and correlated the percent abundance determined in a previous study by 16S metagenomic analysis. Furthermore, authors have sequenced the amplicons obtained by targeting 23S rDNA.

All the protocols and primers used are already reported and established.

The writing style is confusing and contradictory in places that need improvement.

The authors need to establish the article's novelty; This study reports nothing new except that all 4 methods were employed in one study.

The aim of this paper was not to develop a novel, more targeted, experimental strategy. This is an important goal and requires large experimental and bioinformatics efforts, which are indeed currently underway in our laboratory. With the present paper, we wish to address some of the (few) qPCR protocols already published for “Ca. Saccharibacteria”, We test them and compare their “performance”, using combined in vitro and in silico approaches, on samples already subjected to 16S rRNA-based amplicon sequencing. We conclude that none of these available approaches yields a comprehensive and reproducible quantification and taxonomic picture of “Ca. Saccharibacteria”. We believe this adds valuable information, as more and more data points to key roles for “Ca. Saccharibacteria”, and for CPR in general, in important clinical and environmental processes. It is thus conceivable that interest and research on them will increase in the near future. For this reason, a comparative analysis of available protocols could provide valuable information while defining an experimental approach to them. 

During the revision process, we worked on the quality of the paper writing and its readability, also with the help of English-speaking colleagues. Furthermore, we tried to improve readability by simplifying and reorganising some parts, e.g. the Abstract (see below), the Materials and Methods and some parts of Results and Discussion

3. Line 41-43. Rephrase. It is not clear

We have rephrased the sentence.

4. Line 43-48. There are contradictions in claims. Need clarity in writing.

We have rephrased the sentence. In general, we have tried to improve the clarity of the abstract

5. Line 128-129. Give percent abundance of Ca. Saccharibacteria” and unclassified bacteria from the previous study based on 16 S rDNA metagenomics here.

The info requested by the reviewer have been added 

6. Line 162. 16S rRNA is known to have a limited capacity to detect CPR already

Limitations of the study?

The 16S rRNA is the gold standard bacterial marker gene. Even if its limited capacity to detect CPR is known, primers on the 16S gene are normally used for amplicon sequencing. A limitation of this is that >70% of the bacterial clades systematically under-represented or missed in amplicon-based studies belong to CPR, as recently shown by a systematic survey evaluating commonly used 16S “universal” primers (see Eloe-Fadrosh, 2016). Despite this obvious limitation, at the moment there are no well-established alternative marker genes to carry out amplicon sequencing.

Regarding qPCR, three of the four qPCR on “Ca. Saccharibacteria” tested in our paper are designed on the 16S rRNA gene. We believe that testing the performances of these protocols, both bioinformatically and experimentally, could provide useful information for those using this gene in CPR projects . We also tested the only qPCR protocol not designed on the 16S rRNA that we could find at the time of designing the present study.

Eloe-Fadrosh EA, Ivanova NN, Woyke T, Kyrpides NC. Metagenomics uncovers gaps in amplicon-based detection of microbial diversity. Nat Microbiol. 2016 Feb 1;1:15032. doi: 10.1038/nmicrobiol.2015.32. PMID: 27572438.

---

## [Decision Letter · Decision Letter 1]

6 Aug 2024

PONE-D-24-10132R1Comparison of qPCR protocols for quantification of “Candidatus Saccharibacteria”, belonging to the Candidate Phyla Radiation, suggests that 23S rRNA is a better target than 16S rRNAPLOS ONE

Dear Dr. Panelli,

Thank you for submitting your manuscript to PLOS ONE. After careful consideration, we feel that it has merit but does not fully meet PLOS ONE’s publication criteria as it currently stands. Therefore, we invite you to submit a revised version of the manuscript that addresses the points raised during the review process.

As you may find in the attached comments from the reviewers, revision with necessary additional experiments are requested. 

We look forward to receiving your revised manuscript.

Kind regards,

Guanglei Qiu

Academic Editor

PLOS ONE

Journal Requirements:

Reviewers' comments:

Reviewer's Responses to Questions

**Comments to the Author**

1. If the authors have adequately addressed your comments raised in a previous round of review and you feel that this manuscript is now acceptable for publication, you may indicate that here to bypass the “Comments to the Author” section, enter your conflict of interest statement in the “Confidential to Editor” section, and submit your "Accept" recommendation.

Reviewer #1: All comments have been addressed

Reviewer #2: All comments have been addressed

2. Is the manuscript technically sound, and do the data support the conclusions?

Reviewer #1: Partly

Reviewer #2: Yes

3. Has the statistical analysis been performed appropriately and rigorously? 

Reviewer #1: I Don't Know

Reviewer #2: Yes

4. Have the authors made all data underlying the findings in their manuscript fully available?

Reviewer #1: Yes

Reviewer #2: Yes

5. Is the manuscript presented in an intelligible fashion and written in standard English?

Reviewer #1: Yes

Reviewer #2: Yes

6. Review Comments to the Author

Reviewer #1: Authors have responded to all my comments, but refuse to address and do follow up experiment with any of them. They pointed out that they ran out of sample DNA, and recommended comments were too much work for current manuscript. If this article is only repeating what was done before, I am not sure what it is adding to the field. Improving the identification of Saccharibacteria using new qPCR primers will achieve such results.

Reviewer #2: I compared the authors' responses to the comments and am satisfied with the corrections they made to the text of the manuscript.

7. PLOS authors have the option to publish the peer review history of their article (what does this mean?). If published, this will include your full peer review and any attached files.

Reviewer #1: No

Reviewer #2: No

---

## [Author Response · Author response to Decision Letter 1]

3 Sep 2024

JOURNAL REQUIREMENTS:

Journal Requirements:

Done

Comments to the Author

1. If the authors have adequately addressed your comments raised in a previous round of review and you feel that this manuscript is now acceptable for publication, you may indicate that here to bypass the “Comments to the Author” section, enter your conflict of interest statement in the “Confidential to Editor” section, and submit your "Accept" recommendation.

Reviewer #1: All comments have been addressed

Reviewer #2: All comments have been addressed

2. Is the manuscript technically sound, and do the data support the conclusions?

Reviewer #1: Partly

Reviewer #2: Yes

3. Has the statistical analysis been performed appropriately and rigorously?

Reviewer #1: I Don't Know

Reviewer #2: Yes

4. Have the authors made all data underlying the findings in their manuscript fully available?

Reviewer #1: Yes

Reviewer #2: Yes

5. Is the manuscript presented in an intelligible fashion and written in standard English?

Reviewer #1: Yes

Reviewer #2: Yes

6. Review Comments to the Author

Reviewer #1: Authors have responded to all my comments, but refuse to address and do follow up experiment with any of them. They pointed out that they ran out of sample DNA, and recommended comments were too much work for current manuscript. If this article is only repeating what was done before, I am not sure what it is adding to the field. Improving the identification of Saccharibacteria using new qPCR primers will achieve such results.

We assume that this comment refers to point 3 of the original revision, because this is the only point where, in our response, we stated that it was not possible for us to proceed with further experiments with this dataset of DNAs. These samples, in fact, were used for previous work and, physically, many of them are now finished.

In our answer to this point, we explain this. We also briefly discuss why we used the V3-V4 rRNA amplicon sequencing instead of the V1-V3 mentioned by the reviewer. We explain how, in light of recent bioinformatics analysis (for which we cite the relative papers), the V3-V4 16S rRNA amplicon sequencing (and specifically the primer pair we used) performs very well in the context of oral microbiota because it allows optimal OTU clustering at the 97% level, avoiding clustering together sequences belonging to distinct genera. This is a problem that frequently affects several of the most frequently used primers pairs for the description of the oral microbiota. 

Moreover, to further address the point raised by the reviewer, in our first revision we added a sentence in the Conclusions, highlighting that our results were obtained by targeting the V3-V4 regions, but that we could not exclude differences in the results if targeting other V regions.

As explained above, we are not in a position to carry out further experiments. However, to further address the point raised by the referee we now specify in the abstract that our results are based on the V3-V4 regions of the 16S rRNA.

We would also like to add that we disagree with the reviewer, when he/she states that the paper is “only repeating what was done before, I am not sure what it is adding to the field”. This is not the first paper to compare methods, both experimental and bioinformatic. Unlike the reviewer, we have always thought that these are very useful analyses for the community of scientists working on a specific topic. This applies even more to a field such as that of candidate phyla radiation (CPR) and its role in the human microbiota. Research in this field is still in its infancy and with a strong need of understanding “what” the current protocols are able to “see”, both in terms of “who is in there” and “how many”. Naturally, the need for new, more focussed protocols is a pivotal point, to be addressed rigorously, with dedicated work.

Reviewer #2: I compared the authors' responses to the comments and am satisfied with the corrections they made to the text of the manuscript.

7. PLOS authors have the option to publish the peer review history of their article (what does this mean?). If published, this will include your full peer review and any attached files.

Do you want your identity to be public for this peer review? For information about this choice, including consent withdrawal, please see our Privacy Policy.

Reviewer #1: No

Reviewer #2: No

---

## [Editor Report · Decision Letter 2]

5 Sep 2024

Comparison of qPCR protocols for quantification of “Candidatus Saccharibacteria”, belonging to the Candidate Phyla Radiation, suggests that 23S rRNA is a better target than 16S rRNA

PONE-D-24-10132R2

Dear Dr. Panelli,

We’re pleased to inform you that your manuscript has been judged scientifically suitable for publication and will be formally accepted for publication once it meets all outstanding technical requirements.

Kind regards,

Guanglei Qiu

Academic Editor

PLOS ONE
---

## [Editor Report · Acceptance letter]

18 Oct 2024

PONE-D-24-10132R2 

PLOS ONE

Dear Dr. Panelli, 

I'm pleased to inform you that your manuscript has been deemed suitable for publication in PLOS ONE. Congratulations! Your manuscript is now being handed over to our production team.

Kind regards, 

on behalf of

Dr. Guanglei Qiu 

Academic Editor

PLOS ONE